**PLOS | ONE**

# Fragmentation and low density as major conservation challenges for the southernmost populations of the European wildcat

Jose María Gil-Sánchez[1]*, Jose Miguel Barea-Azcón[2], Javier Jaramillo[2], F. Javier Herrera-Sánchez[3], José Jiménez[4], Emilio Virgós[5]

**1** Department of Zoology, Universidad de Granada, Granada, Spain, **2** Agencia de Medio Ambiente y Agua (Consejería de Agricultura, Ganadería, Pesca y Desarrollo Sostenible, Junta de Andalucía), Gerencia de Granada, Edificio Zeus III, Granada, Spain, **3** Harmusch, Wildlife Research and Conservation,Ubrique, Spain, **4** Instituto de Investigación en Recursos Cinegéticos (IREC, CSIC-UCLM-JCCM), Ciudad Real, Spain, **5** Department of Biology, Geology, Physics and Inorganic Chemistry, ESCET, Universidad Rey Juan Carlos, C/Tulipán, Móstoles, Madrid, Spain

* jmgilsanchez@yahoo.es

**Data Availability Statement:** All relevant data are within the paper and its Supporting Information files.

## Abstract

Knowledge of population dynamics of threatened species in the wild is key to effective conservation actions. However, at present, there are many examples of endangered animals for which their current situation is unknown, and not just in remote areas and less developed countries. We have explored this topic by studying the paradigmatic case of the European wildcat (*Felis silvestris silvestris*), an endangered small carnivore whose status has been subjectively established on the basis of non-systematic approaches and opportunistic records. Little is known about its demographic situation, prompting the need for information to improve conservation measures. However, the secretive behaviour of felines along with its low density in natural conditions have prevented the gathering of sufficient data. We developed a field sampling strategy for one of the largest populations (Andalusia, South Spain, 87,268 km$^2$), based on a logistically viable systematic non-intrusive survey by camera-trapping. This study offers the first large-scale estimation for any European wildcat population, based on analytical approaches applied on Species Distribution Models. A hierarchical approach based on a Maxent model for distribution estimation was used, along with Generalised Linear Models for density estimation from explicit spatial capture-recapture data. Our results show that the distribution range is smaller and more highly fragmented than previously assumed. The overall estimated density was very low (0.069 ±0.0019 wildcats/km$^2$) and the protected areas network seems to be insufficient to cover a significant part of the population or a viable nucleus in demographic terms. Indeed, the most important areas remain unprotected. Our main recommendations are to improve the protected area network and/or vigilance programs in hunting estates, in addition to studying and improving connectivity between the main population patches.

**Funding:** The research was partially funded by the Consejería de Medio Ambiente y Ordenación del Territorio (www.juntadeandalucia.es > medioambiente > site > portalweb) through the European Union (FEDER Project http://www.juntadeandalucia.es/medioambiente/site/portalweb/menuitem.6ffc7f4a4459b86a1daa5c105510e1c/?vgnextoid=05cf8706a8bb9510VgnVCM100000132 5e50aRCRD&vgnextchannel=05cf8706a8bb9510 VgnVCM1000001325e50aRCRD) and is part of the Global Change Observatory of Sierra Nevada (https://digibug.ugr.es/handle/10481/54686). J.M. G.-S. was supported by a Prometeo fellowship from the SENESCYT and the national agency for Education and Science of the Government of Ecuador (https://www.educacionsuperior.gob.ec/prometeo/). There aren't specific grant numbers or funding from commercial companies. The funders had no role in study design, data collection and analysis, decision to publish, or preparation of the manuscript.

**Competing interests:** The authors have declared that no competing interests exist.

# Introduction

Knowledge of the population dynamics of threatened species in the wild is key to effective conservation actions [1, 2]. While this is an obvious idea, at present, there are many examples of endangered animals for which their current situation is unknown, and not only in remote areas and less developed countries. This is the paradigmatic case of the European wildcat (*Felis silvestris silvestris*), an emblematic feline that has been the target of several studies on its ecology (e.g. [3, 4, 5, 6, 7, 8, 9, 10, 11, 12, 13, 14, 15]) and especially, on its problematic hybridization with domestic cats [16, 17, 18, 19, 20, 21, 22, 23, 24, 25, 26, 27]. Today, it is assumed to be an endangered taxa in most of the countries in which it lives [28]. It is a legally protected species in Europe, both through the Bern Convention and the European Habitats Directive, which consider the wildcat as "strictly protected". The available distribution maps show a severely fragmented range, with the main patches in the Iberian Peninsula, France, Germany and the eastern countries of Europe (see a compilation in [28]). However, most of these maps (if not all) have been subjectively built on the basis of non-systematic approaches, using opportunistic records which could be assumed to be unchecked following the detailed phenotypic examination needed for hybridization detection [29, 30, 31]. In fact, these kind of data have traditionally been the sole source of information to define the wildcat's conservation status (see e.g. the case of Spain, where one of the largest populations survives; [32]). Hence, we must ask what is the current situation of the European wildcat populations? More simply, how many "pure" European wildcats are left and how many live under the umbrella of the protected areas network, such as in national parks or other reserves? A paradoxical situation has arisen in which the ecology of the subspecies is well-known, due to ample research on this topic, while very little is known about the species' demographic situation; at the same time, answers to the latter question are critical for the effective conservation of these endangered European wildcats. However, the secretive behaviour of felines along with its low densities in natural conditions [33] have prevented answers to these questions until now. A large-scale conservation strategy for a species must take into account the complexity of density-niche relationships [34] and simultaneously overcome the huge challenges posed by density estimations over large spatial scales (e.g. [35]).

Fortunately, today there is a set of useful methods available to both calculate densities of elusive species [36] and to predict potential distribution and density estimates over large regions based on suitability indexes derived from niche-species models [34]. The use of solid density estimates and robust predictions of suitable areas for the distribution of species allow managers to take well-informed conservation decisions over large spatial scales with reduced field and economic effort. Furthermore, combinations of these methods are also useful to develop long-term monitoring programs, a key element for any strategic conservation programme of an endangered species [37, 38].

For abundance estimations, non-intrusive capture-recapture approaches based on camera-trapping or molecular identification of scats and hairs [39] have proven highly efficient for felines (e.g. [39, 40, 41]), including in the case of the rare European wildcat [12, 42, 43, 44, 45, 46, 47]. On the other hand, during the last decades, species distribution models (hereafter SDMs) have been developed to provide solutions to a wide range of ecological, biogeographical and conservation problems (reviewed in [48]). Statistical methods to calculate distribution ranges and suitability indexes of abundance from partially known occurrence data are available (e.g. [34, 49, 50]). SDMs are especially robust when a good dataset of environmental predictors is available, as has been established for our target species in Germany [51] or Portugal [11] and for other felines elsewhere [52, 53, 54, 55]. Finally, non-intrusive surveys of wildcats have proven to be optimal approaches to estimate the degree of genetic introgression of the domestic cat [43, 46, 47].

The present study aims to evaluate the use of these combined sets of field and analytical methods, to carry out large-scale spatial surveys on the status of European wildcat populations, as a practical example of the demographic diagnosis of felines by non-intrusive surveys that can be useful elsewhere. We have developed a field sampling strategy of one of the largest populations of the species, located in mid-southern Spain in the Andalusia region, where a logistically viable survey was designed to be carried out by practitioners and for long-term monitoring. From the resulting field data, our goals were: (1) to estimate the entire population of wildcats in Andalusia; (2) to estimate domestic (feral) cat abundance in the wild in order to approximate the hybridization problem; and (3) to evaluate the contribution of the protected areas network to European wildcat conservation.

## Materials and Methods

### Study area

The study was carried out in Andalusia (87,268 km$^2$), southern Spain (Fig 1). Andalusia is a typical Mediterranean area made up of three main regions (Fig 1), from north to south: (1) *Sierra Morena*, is a low mountain range (altitude 50–1298 m a.s.l.) dominated by well-preserved Mediterranean oak forests (*Quercus ilex*, *Q. suber*, *Q. faginea*) and scrublands (*Arbutus unedo*, *Phylliera altifolia*, *Cistus ladanifer*, *Lavandula stoechas*), and some pine plantations (*Pinus pinea*, *P. pinaster*). The main uses of land are big game (*Cervus elaphus*, *Sus scrofa*) and cattle, with a relatively low human density. It is one of the best-preserved areas in Europe, holding the largest population of the endangered Iberian lynx (*Lynx pardinus*). (2) *Guadalquivir River valley*, is a low-land region (0–500 m a.s.l.) transformed by cultivation, mainly of olive trees, sunflower and cereal crops and is densely populated by humans. The only remaining patch of wild landscape in this area is the *Doñana* plain (Fig 1), which is very important for wildlife conservation at an international level. (3) The *Sierras Béticas* are a complex mountain system (0–3478 m a.s.l.) where the impact of crops (olive and almond trees, cereal) and cattle has led to severe degradation of natural vegetation; the autochthonous forests and scrublands are fragmented (Fig 1), although they hold one of the highest levels of botanical diversity and uniqueness in Europe [56], from moist cork oak forests in the west to arid sub-desert scrubs in the east, with autochthonous boreal relicts of pine forests (*P. sylvestris*) in the highest mountains. Human density is spatially variable but relatively high on average.

The wildcat is assumed to be well distributed along the Sierra Morena and the Sierras Béticas, a range based on opportunistic unconfirmed records [32]. Twenty-four Natural Parks and two National Parks cover 17.58% of the study area (15.338,88 km$^2$), providing protection to most of Doñana, and patches of the Sierra Morena and Sierras Béticas ranges (Fig 1).

### Field surveys

Three types of non-intrusive field surveys have been successfully developed for the wildcat: camera-trapping [44, 45, 46, 47], scat sampling for molecular identification [46] and hair sampling, also for genetics [42]. We used the first two methods, while ongoing field studies by our team are showing a lesser efficiency of hair traps in our study area. Twenty-two survey blocks (Fig 1) were distributed in Sierra Morena (10 blocks) and Sierras Béticas (12 blocks) between 2011 and 2015; this distribution was biased towards eastern Andalusia mainly because the habitats there are much more heterogeneous and representative of the other areas. All sampling blocks were selected first assuming that they were included in the potential habitats of the wildcat (see study area), and second, attempting to represent the main forest and scrubland types of the overall potential range within Andalusia. Third, limitations of access facilities and/ or authorizations for private lands were considered.

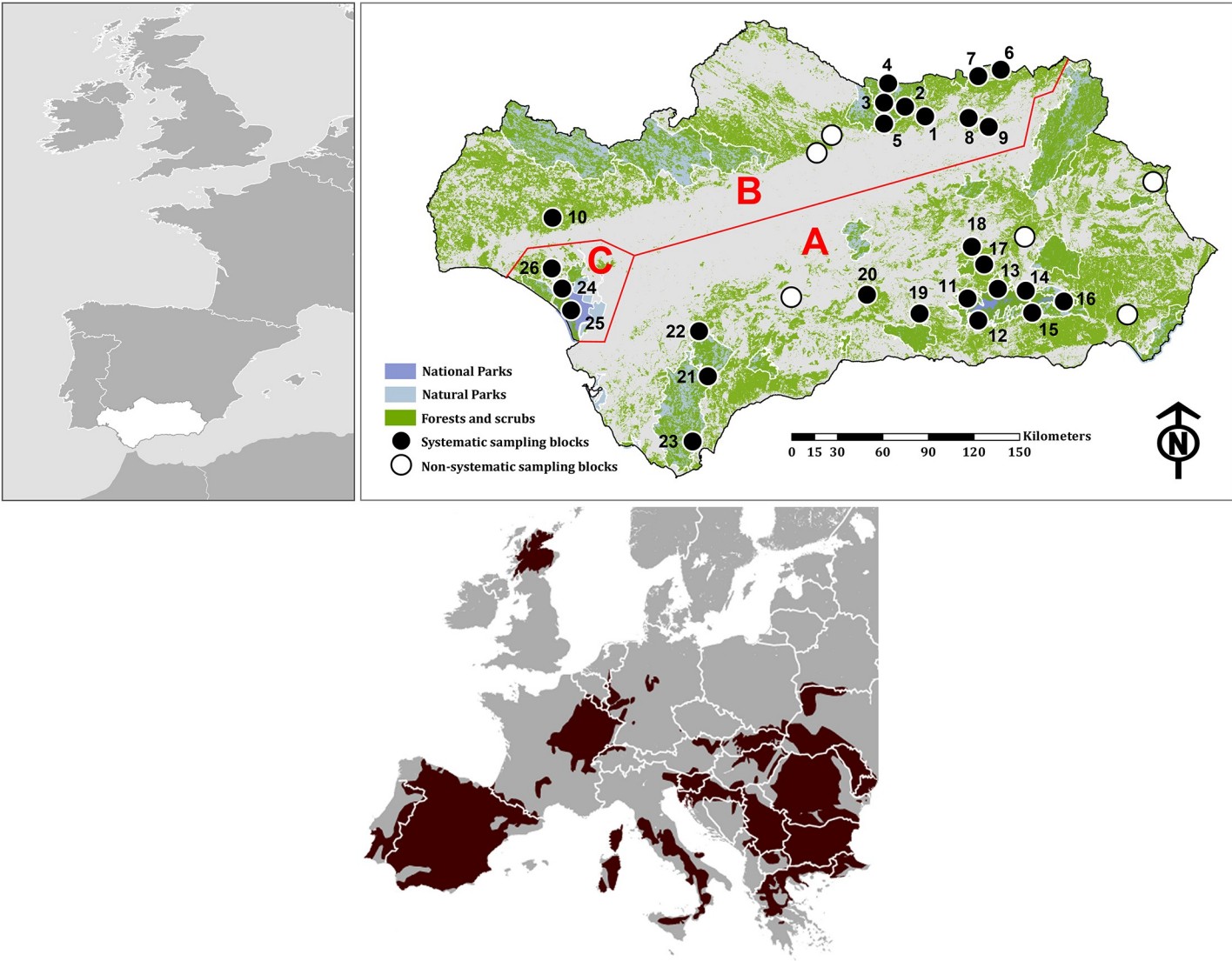

**Fig 1. Study area showing the distribution of the camera-trapping blocks (dots) within Andalusia and the distribution range assumed for the European wildcat** [94]. Geographic areas considered in the present study: A Sierras Béticas, B Sierra Morena, C Doñana. Block 23 was carried out by Gómez-Chicano et al. [57]; and blocks 24, 25 and 26 by Soto and Palomares [43].

For each block, we designed a conventional camera-trapping survey, consisting of the installation of 8–12 camera traps in most cases (Table 1) following the procedure described by Gil-Sánchez et al. [47]. Lures were used in all the blocks, both live pigeons in cages (12 blocks) and lynx urine (9 blocks, Table 1). The urine was employed within areas of high risk of robbery and vandalism, because this type of lure makes it easy to camouflage the cameras. Lynx urine was found to be slightly less efficient for wildcats compared to pigeon lures [47], and therefore we assumed lower effects on the comparative results. The cameras were infrared-triggered (DLC Covert™, Leaf River™ model IR-5 and Scout Guard™ model SG565F-BM), with a sampling period for each camera ranging from two to three months. To increase the field data set, we used the results of seven camera-trapping surveys carried out by other teams and data from two published surveys ([43, 57]; see Fig 1).

**Table 1. Details of the 26 systematic camera-trapping blocks (see Fig 1 for location of each).** Grey rows: sampling blocks with SCR calculations (N = 11, see details in Table 2); lure: (p) pigeon bait, (o) lynx urine, *data from Simón et al. [58].

| #Block | Coordinates | Camera stations (lure) | Camera days | Wildcat | | | | Domestic cat | | Hybrid cat | | Walked km | Cat scats | Rabbit (latrines/ km) |
|---|---|---|---|---|---|---|---|---|---|---|---|---|---|---|
| | | | | Individ. | Captures | Cap/ 100 cam-day | D (ind./ 100 km²) | individ. | captures | Individ. | captures | | | |
| 1 | 38.168266–3.959968 | 14 (p) | 1333 | 4 | 7 | 0.52 | 6.59 | 0 | 0 | 0 | 0 | 10.9 | 0 | 19.08 |
| 2 | 38.175024–4.010570 | 16 (p) | 924 | 3 | 3 | 0.32 | 4.68 | 0 | 0 | 0 | 0 | 17.3 | 0 | 14.68 |
| 3 | 38.229479–4.167551 | 9 (p) | 769 | 4 | 14 | 1.82 | 6.55 | 0 | 0 | 0 | 0 | 9.9 | 0 | 5.03 |
| 4 | 38.305422–4.187788 | 5 (p) | 335 | 2 | 5 | 1.49 | 5.95 | 0 | 0 | 0 | 0 | 7.5 | 0 | 7.38 |
| 5 | 38.182323–4.156388 | 5 (p) | 336 | 0 | 0 | 0 | 0 | 0 | 0 | 0 | 0 | 5.2 | 0 | 0 |
| 6 | 38.491827 -3-264204 | 12 (p) | 998 | 2 | 17 | 1.70 | 4.90 | 0 | 0 | 0 | 0 | 5.4 | 0 | 25.9 |
| 7 | 38.471045–3.448069 | 4 (o) | 1288 | 3 | 9 | 0.69 | 4.49 | 0 | 0 | 0 | 0 | 9.2 | 1 | 35,76 |
| 8 | 38.162388–3.570727 | 4 (o) | 180 | 2 | 16 | 8.88 | 19.48 | 0 | 0 | 0 | 0 | 14.4 | 3 | 34,48 |
| 9 | 38.161735 -3-512425 | 4 (o) | 148 | 2 | 2 | 1.35 | 5.70 | 1 | 1 | 0 | 0 | - | - | - |
| 10 | 37.479996–6.658689 | 10 (o) | 180 | 0 | 0 | 0 | 0 | 0 | 0 | 0 | 0 | - | - | 0* |
| 11 | 37.097731–3,494577 | 10 (p) | 686 | 1 | 1 | 0.14 | 2.67 | 0 | 0 | 0 | 0 | 20.1 | 2 | 0 |
| 12 | 36.960573–3.407763 | 12 (p) | 840 | 2 | 3 | 0.35 | 4.25 | 2 | 2 | 0 | 0 | 10.5 | 0 | 0 |
| 13 | 37.191936–3.251752 | 9 (p) | 544 | 1 | 1 | 0.18 | 3.18 | 0 | 0 | 0 | 0 | 12.8 | 0 | 0 |
| 14 | 37.125307–3.075088 | 10 (p) | 686 | 1 | 1 | 0.14 | 2.56 | 0 | 0 | 0 | 0 | 23.7 | 0 | 0,97 |
| 15 | 37.063330–2.995206 | 11 (p) | 763 | 3 | 10 | 1.31 | 6.11 | 0 | 0 | 0 | 0 | 12.8 | 0 | 0.078 |
| 16 | 37.084784–2.766284 | 12 (p) | 759 | 5 | 37 | 4.87 | 10.46 | 0 | 0 | 0 | 0 | 12.2 | 0 | 0.32 |
| 17 | 37.296489–3.401067 | 8 (o) | 457 | 1 | 2 | 0.43 | 4.01 | 0 | 0 | 0 | 0 | 51.5 | 0 | 3.66 |
| 18 | 37.377995–3.417979 | 16 (o) | 450 | 7 | 32 | 7.11 | 17.55 | 0 | 0 | 0 | 0 | 74.1 | 2 | 29,29 |
| 19 | 36.997621–3.817692 | 7 (o) | 200 | 1 | 1 | 0.50 | 4.14 | 0 | 0 | 0 | 0 | 24.6 | 0 | 13.26 |
| 20 | 37.021747–4.217015 | 8 (o) | 351 | 0 | 0 | 0 | 0 | 2 | 5 | 0 | 0 | 5.0 | 1 | 39.5 |
| 21 | 36.499840–5.460368 | 10 (o) | 647 | 0 | 0 | 0 | 0 | 0 | 0 | 0 | 0 | 6.5 | 0 | 0.1 |
| 22 | 36.792605–5.397782 | 12 (o) | 1020 | 0 | 0 | 0 | 0 | 0 | 0 | 0 | 0 | 108.7 | 0 | 0.05 |
| 23 | 36.242679–5.609176 | 7 (p) | - | 0 | 0 | 0 | 0 | 0 | 0 | 0 | 0 | - | - | 0* |
| 24 | 37.146208–6.551801 | 124 (p,o) | 4329 | 4 | 23 | 0.53 | 4.19 | 2 | 2 | 3 | 3 | - | - | 5.83* |

(Continued)

**Table 1.** (Continued)

| #Block | Coordinates | Camera stations (lure) | Camera days | Wildcat | | | | Domestic cat | | Hybrid cat | | Walked km | Cat scats | Rabbit (latrines/ km) |
|---|---|---|---|---|---|---|---|---|---|---|---|---|---|---|
| | | | | Individ. | Captures | Cap/ 100 cam- day | $D$ (ind./ 100 km$^2$) | individ. | captures | Individ. | captures | | | |
| 25 | 37.004890– 6.505503 | 35 (p,o) | 1190 | 0 | 0 | 0 | 0 | 0 | 0 | 0 | 0 | - | - | 6.73* |
| 26 | 36.964384– 6.450216 | 7 (p,o) | 242 | 2 | 5 | 2.06 | 6.99 | 0 | 0 | 0 | 0 | | | 10.33 |

For most of the sampling blocks (n = 19, Table 1) and simultaneously for the remote camera surveys, walking surveys of scats were designed following the protocols of Anile et al. [46]. However, after an effort of 442.3 walked km (see Table 1) carried out by well-trained personnel (J. M. Gil-Sánchez; J. Herrera-Sánchez), only nine putative wildcat scats were found. Therefore, this method was inefficient for our study area. The walking surveys were used to sample rabbit latrines, as a method to estimate the abundance of this key prey species for wildcat in the Iberian Peninsula [10, 13]. The sampling period lasted from 2011 to 2015.

## Identification of cats

We identified each individual as a domestic cat, wildcat or as a hybrid cat after a detailed examination of the coat patterns and the shape of the tail [29, 30, 45, 47]. Seventeen wildcats of our study area were genetically examined in order to detect hybridization with domestic cat [30]; we found that the phenotypic traits of genetically pure individuals of our study area were very constant, and we only considered typical wildcats from camera trapping as "pure" individuals (most of them identical to WC1 in Fig 2 of [30]; see some pictures from camera traps in [47]). Camera traps detected three putative hybrid cats (see Results), which had coat pattern and tail shape were very close to typical wildcats, but had wide white patches on the pelage.

## Environmental data

To model the distribution and abundance of the wildcat in the study area, we incorporated 54 environmental descriptors attending to five conceptual groups (climatic, relief, vegetation, water availability and human presence) of environmental variables selected to represent different resources. All the variables were obtained at 40-metre resolution. Previous to the modelling approach, we evaluated Pearson correlations among these selected independent variables to avoid multicolinearity in the models [59]. We chose those with the greatest biological significance for wildcats based on our expertise and on the habitat preferences previously described for this species [7, 11, 13, 51]. We carried out the same exploration, looking for the variables statistically related to rabbit abundance that were evaluated in the systematic blocks. As a result, we obtained a list of 25 uncorrelated environmental predictors (Table 2). The unique climatic variable was cumulative rainfall. To obtain annual averages of rainfall at a 40-metre resolution we applied the climate mapping method proposed by Ninyerola et al. [60], taking as an input the daily records of 1000 weather stations contained in the Andalusian Information Subsystem for Environmental Climatology. Topographic and water availability-related variables were calculated from a 40 meter resolution terrain elevation model provided by the Environmental Information Network of Andalusia (REDIAM, Andalusia Government). The elevation model was then processed through GRASS GIS software (GRASS Development Team, 2009) using R.PARAM.SCALE, R.SLOPE.ASPECT, R.TERRAFLOW, R.SUN and R.

**Table 2. Estimates of relative contributions of the environmental variables to the MaxEnt model.**

| Variable | Percent contribution | Permutation importance |
|---|---|---|
| Frequency of cultivated areas | 16.8 | 17.5 |
| Frequency of oak forests | 11.1 | 3.4 |
| Frequency of urban areas | 9.7 | 17.9 |
| Distance to water bodies | 9.5 | 2 |
| Precipitation (accumulated) | 5.8 | 4.4 |
| Frequency of pine forests | 4.6 | 1.7 |
| Frequency of forest | 3.7 | 1.7 |
| Frequency of eucalyptus plantations | 1.9 | 1.6 |
| NDVI (Normalized Difference Vegetation Index) | 1.8 | 2.6 |
| Frequency of dense scrubland | 2.7 | 1.7 |
| Frequency of pasturelands | 4.3 | 3.6 |
| Elevation | 4.3 | 13.3 |
| Convergence index | 0.7 | 0.6 |
| Hour of sun during winter | 1.5 | 0.6 |
| Medium solar radiation during winter | 1.5 | 0.1 |
| Frequency of olive cultivations | 1.5 | 7.3 |
| Topographic exposure | 0.6 | 1.8 |
| Slope | 3.2 | 1.8 |
| Distance to roads | 3.2 | 4.3 |
| Medium solar radiation during summer | 3.2 | 0.5 |
| Frequency of dispersed scrubland | 4 | 4.2 |
| Frequency of water bodies | 2 | 3.1 |
| Solar radiation during summer | 0.2 | 0.1 |
| South-North gradient | 1 | 2.2 |
| East-West gradient | 1 | 2.1 |

RECODE modules. Land cover or land use variables were obtained from the land cover and land use map of Andalusia (SIOSE Andalusia, year 2003, scale 1:25:000). These vector maps were then transformed into raster maps and the distance to target entities were calculated using V.EXTRACT, V.TO.RAST and R.GROW.DISTANCE GRASS GIS modules. Frequencies were also calculated from these rasters using a neighborhood analysis through R.NEIGH-BORS GRASS GIS module. The result was the number of pixels with a presence of a given entity within a 1000-metre radius.

## Niche models based on presence-only data

To model the distribution of the wildcat in Andalusia, we first selected a dataset of independent samples from camera-trapping. We only used locations separated by at least 1887 metres (n = 68). This distance was the average wildcat home range radius in Iberian Mediterranean ecosystems of the southern Iberian Peninsula [11]. SDMs were performed using MaxEnt (version 3.4.1k; [61, 62]), after checking recommendations by Merow et al. [63] and Yackulic et al. [64]. MaxEnt provides SDMs from presence-only species records and shows good predictive performance when the presence dataset sample is low in comparison to other modelling algorithms, as it was in our case [65]. MaxEnt models were generated, after 500 iterations, with the dataset of 68 presence records. The final result of the MaxEnt model was a continuous map that was transformed into binary using a cutoff point where sensitivity equals specificity. This threshold probability was 0.262. Finally, we removed potential habitat patches of less than 228

hectares, equivalent to the minimum female wildcat home range described within a southern Iberian Mediterranean ecosystem [11].

Our model performance was evaluated using a receiver operating characteristic (ROC) curve. From this curve, the area under the ROC curve value (AUC) is a widely-accepted method to evaluate SDM performance (e.g. [65]). The MaxEnt output was re-evaluated by comparing the predictive map with radio-tracking data (see [51]). We used 370 independent locations of nine resident radio-tagged wildcats (four adult females and five adult males), which were captured within a camera-trapping block (#18, Fig 1) in the Béticas range; radio-tracking periods were March 2003 to September 2004 and November 2017 to February 2019. Following a scientific standardized protocol designed and largely used for our target species [15], animals were captured with box-traps (metal cages of 100 x 50 x 70 cm, porting in our case a wooden roof to prevent from sun or rains), using live house pigeons (*Columba* sp.) as lure, unavailable to captured carnivores thanks to an isolation cage that prevents injuries. The pigeons were released at the end of the trapping sessions. Box-traps were checked daily after sunrise and before sunset, in order to minimize animal stress and to supply food and water to pigeons. Alternatives capture methods for wildcats (e.g. leg-hold traps or snare traps) have a large risk to fatal injuries and, therefore, were rejected. Following the wildlife laws of Spain (which include any ethic consideration), this research was approved by the regional environmental authorities (Consejería de Medio Ambiente y Ordenación del Territorio, Dpto. Geodiversidad y Biodiversidad, approval number: 201699900550733). Once the cats were captured, they were immobilized by veterinarians using an anesthetic (Xylazine and ketamine hydrochloride) at a dose of 10 mg kg$^{-1}$. To evaluate the accuracy of MaxEnt, we explored the lineal distance to the nearest predicted patch by pooling it into five categories: inside optimal patch (0 m.), at <250 m, at <500 m, at <1000 m and at >1000 m, carrying out a Chi-square test to evaluate if the observed frequency distribution was different from the null distribution.

## Density estimations of wildcats

Only adult or sub-adult individuals were taken into account to avoid seasonal effects. Once the taxonomic status was established, each cat was individually identified following the protocol of Anile et al. [45]. We then carried out density estimation within each sampling block by using spatially explicit capture-recapture (SCR) models, that are thinned spatial point process models used to make inferences about the abundance and distribution of animal activity centres [66, 67]. SCR models allow for inference about population size and density by modelling capture probability as a function of the distance between activity centres and detectors (e.g. camera-traps). The SCR capture probability function typically includes two main parameters: the scale parameter of the half-normal distribution (sigma), which is determined by home range size; and the baseline detection rate, that is the probability of encountering an individual at its activity centre. In order to improve parameter estimates when sample sizes (spatial recaptures) were small [68, 69, 70], we used models in a Bayesian approach sharing among sites sigma and baseline detection parameters. The models were fitted using a script written in Nimble [71, 72] and R [73]. Three parallel Markov chains with 100.000 iterations each (burn-in = 1000 iterations, thinning rate = 1) were run. The Gelman–Rubin statistic, R-hat [74], was used for checking chain convergence, which compares between and within chain variation [75]. R-hat values below 1.1 indicate convergence.

We carried out the SCR calculations for eleven blocks holding more than nine camera stations (Table 1); with them, we carried out a regression analysis between density estimations (individuals /km$^2$) and relative abundance (captures/100 camera-days) with the goal of obtaining a formula for transforming to density the relative abundance of the rest of the blocks [76, 77].

## Population size-niche predictor relationships

We used generalised linear models (GLM) where the response variable was the density of wild-cats. In these models, we used as explanatory variables the 25 variables cited above (Table 2). These variables were quantified within a 3-km circle centred at the centre of each remote cam-era-sampling block, resulting in a buffer that included the whole minor convex polygon of every block. We also excluded the four blocks with the highest presence of Iberian lynx (#1, #2, #3 and #9, Table 1; data from our survey) since strong competitive exclusion was expected [58] independently of the environmental descriptors, which hampered the accuracy of the results as we confirmed in early GLM calculations. We carried out a GLM analysis with normal error distribution (confirmed by Kolmogorov-Smirnov tests and q-q plots) and identity link func-tion. For model selection, we first selected explanatory variables with significant associations with the response variable in univariate tests ($R_p$ correlation). To find the best model explain-ing wildcat density we used a multi-model selection approach where the importance of vari-ables and the values of estimate coefficients were averaged across similarly supported models [78]. In brief, we evaluated all combinations of predictors and models with different levels of complexity. We selected only the models with AICc values lower than two in relation to the best model (lower value of AICc). We also computed the relative importance of the variables from their Akaike weights ($W_i$) and the average values of the estimated coefficients and their standard errors [78]. To analyse the model fit, we calculated the R-squared of the final model. We carried out the statistical analyses with R software version 3.4.2 [73] using the package MuMIn [79] for multi-model selection.

## Population size estimation and coverage of protected areas

The best GLM model was resampled from a 40-metre resolution raster to a UTM 5x5 km square net (using the spatial analysis extension on ArcMap 10). The 5x5 UTM square is a geo-graphic unit similar in size (25 km$^2$) to the average camera sampling circle (28.2km$^2$), and thus it has a remarkable biological significance for European wildcat spatial ecology (range of terri-tory in southern Mediterranean Iberian Peninsula = 1.70–13.71km$^2$ [11, 15]). We overlapped the 5x5 km square net with the resulting MaxEnt wildcat distribution map of Andalusia, and then removed the 5x5 km squares with less than 10% of potential presence ($<2.5$ km$^2$), since they did not reach the minimum size for a female wildcat territory (2.28 km$^2$ [11]). The popu-lation size $N$ (mean, standard error and 95% of confidence interval) was calculated from the spatial estimate values of density as: $N = (\sum d_i/d)^* S$, with $d_i$ being the density of each 5x5 km square, $d$ the total number of squares and $S$ the total range size (km$^2$) derived from the MaxEnt presence surface within the 5x5 km squares.

To estimate the wildcat population covered by each National or Natural park (hereafter Natural Protected Areas or NPAs), we carried out the same calculations previously described. For our analyses, we only considered UTM 5x5 with at least 75% of its area included in the NPA.

## Results

### Distribution range inference

The MaxEnt model shows a very high predictive performance, in that the training AUC was 0.96. Thus, the model can be considered as potentially useful (see ROC curve in S1 Fig). The most important environmental predictor was agricultural lands frequency with a negative response curve, followed by the frequency of oak forests with a positive response, the frequency of urban areas with a negative response, the distance to water bodies with a positive response

and altitude with a positive (but partially semi-quadratic) response (Table 2; S2 Fig). The rest of the predictors showed low contribution ranging from 0.1–5.8% of contribution and permutation importance (Table 2; see response curves in S1 Fig and Jackknife test in S3 Fig).

The surface defined by our model shows a potential distribution for the European wildcat in the study area of 8558.73 km². This area is distributed in 476 patches with an average size of 15.89 km² (range 2.28–5651.14 km²). Eighty percent of the total area is concentrated in 7.8% of the largest patches and this implies that the majority of the distribution area of the European wildcat in Andalusia is restricted to 37 localities. The distribution model results showed two main populations (Fig 2): the largest one with a continuous distribution at Sierra Morena (4652.87 km²), and another largely fragmented one at Sierras Béticas (3730.61 km²), where the main optimal patches were located in the western mountains. Doñana (175.24 km²) appeared as a secondary and somewhat isolated optimal area, but spatially related to Sierra Morena (Fig 2).

A percentage of 48.1 of the radio-tracking locations fell within the predicted range, 24.8% at <250 m, 11.6% at <500 m, 7.0% at <1000 m and 8.1% at >1000 m. The Chi-square test showed that this observed frequency distribution was different from the null distribution (Chi-square = 85.9; $P<0.00001$). The average lineal distance to the nearest predicted patch was 323.5 m ($ES$ = 40.9 m). Only the home range of one male in 2003–2004 fell outside of the predicted range; this home range was unoccupied during 2018 (J.M. Gil-Sánchez data from intensive camera-trapping).

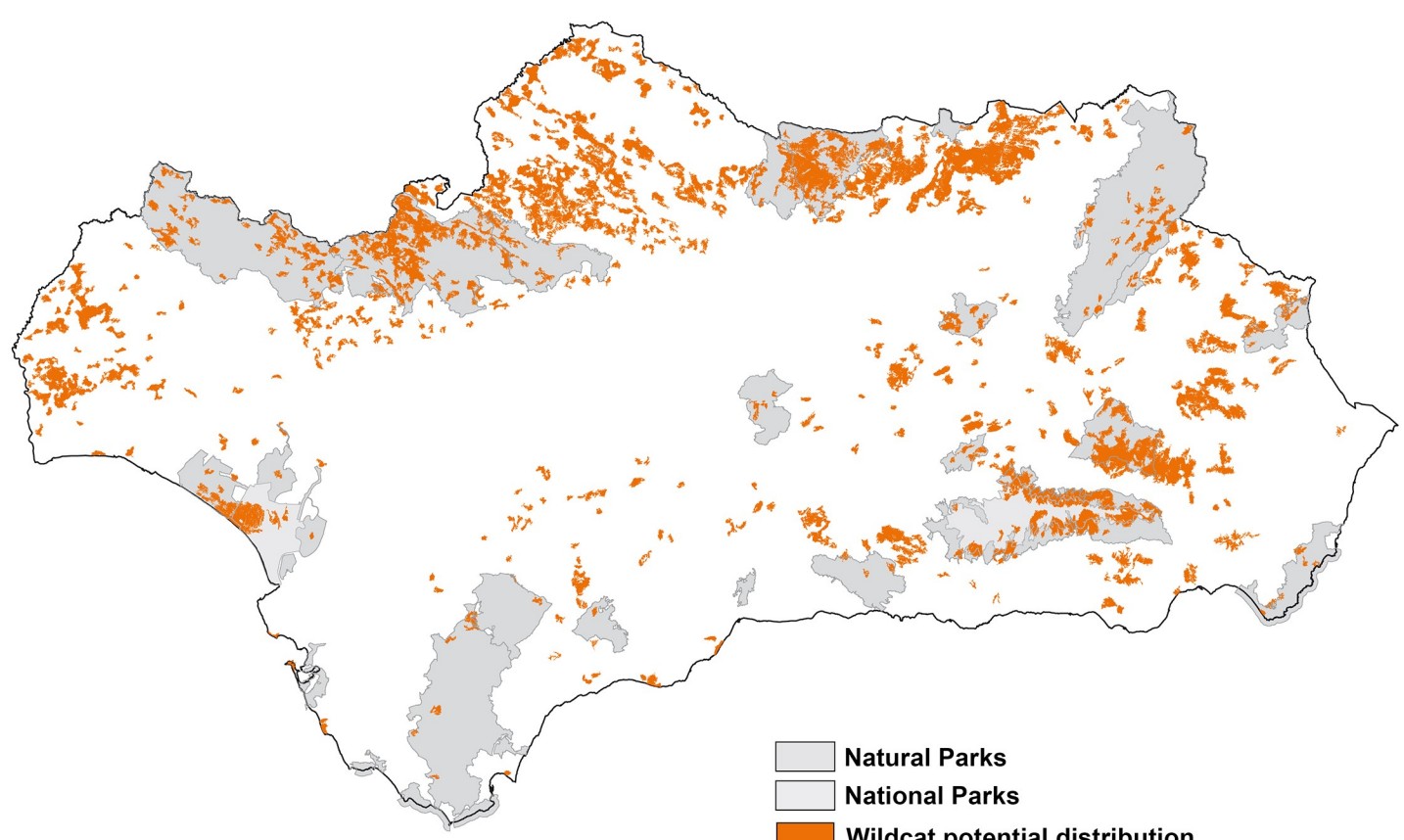

**Fig 2. Wildcat potential distribution in the study area modelled with MaxEnt (patches of more than 228 hectares, see text for further details).**

## Density estimations of wild-living cats

Forty-four wildcats were captured on 189 occasions at 19 systematic blocks (Table 1). SECR calculations showed a wide range of densities, from 0.02 to 0.17 wildcats/km$^2$, although low densities were the most frequent (Table 3). We found a significant relationship ($R_s$ = 0.92, $P$ = 0.0001) between $D$ and the capture rate for the eleven available blocks with $D$ estimation. Therefore, we used this lineal regression formula (wildcats/100 km$^2$ = (1.83*captures/100 camera-days) + 3.23) to estimate $D$ for the rest of the blocks (Table 1). The captures of wild-living domestic cats (both feral cats *sensu stricto* and roaming house cats) did not allow for density calculations; there were only seven individuals with nine captures (Table 1). They were detected in 14.8% of the sampling blocks. Only four putative hybrids were detected in four blocks, three of them in Doñana (see pictures in [43]) and another in a non-systematic block in Sierra Morena. The capture rates (individuals/100 camera days) were 1.39 for wildcats, 0.098 for domestic cats and 0.029 for putative hybrids (2.04% of apparent hybridization rate).

## Population size modelling

We only detected two significant predictors with wildcat density: olive crop cover ($R_p$ = 0.66, $P$ = 0.002) and precipitation ($Rp$ = -0.49, $P$ = 0.049). Rabbit abundance index (latrines/km) showed a positive relationship with wildcat density ($R_p$ = 0.54, $P$ = 0.009) and olive crops ($R_p$ = 0.74, $P$ = 0.0001), and a negative relationship with precipitation ($R_p$ = -0.39, $P$ = 0.032). The multi-model GLM performed with the two predictors generated a final model including both variables, although olive crop cover showed higher relative importance than precipitation (1 vs 0.32). The multimodel approach laid two equally probable models, one including only olive crop cover and another including the two predictors. The model with only olive crop cover showed a lower AICc value, and higher Akaike Weight (Table 4). Adjusted R-squared values indicated a good fit of the model including both variables, which explained 59.73% of the total variance of wildcat density. Parameter estimates of the full-averaged coefficients of the model can be seen in Table 4.

The distribution of the regional wildcat abundance is shown in Fig 3, where three core areas can be observed: central-eastern Sierra Morena, central-western Sierra Morena and north-eastern Sierras Béticas, whereas the rest of the range usually holds low or very low densities. Total estimation was near one thousand individuals, with more than five hundred in

**Table 3. Density estimations (individual/km$^2$) by Bayesian Spatial Explicit Capture Recapture models (block #18 to #14).** See Table 1 for details of each sampling block. $\lambda_0$ is the baseline detection rate, and σ the parameter of scale from the half-normal distribution, related to the home range.

| # block | mean | sd | CV | Quantiles | | |
|---|---|---|---|---|---|---|
| | | | | 2.50% | 50% | 97.50% |
| 18 | 0.1755 | 0.0677 | 0.39 | 0.0745 | 0.1663 | 0.3383 |
| 16 | 0.1046 | 0.0437 | 0.42 | 0.0438 | 0.0938 | 0.2126 |
| 1 | 0.0659 | 0.0290 | 0.44 | 0.0270 | 0.0595 | 0.1351 |
| 3 | 0.0655 | 0.0288 | 0.44 | 0.0243 | 0.0582 | 0.1359 |
| 15 | 0.0611 | 0.0301 | 0.49 | 0.0226 | 0.0566 | 0.1359 |
| 6 | 0.0490 | 0.0286 | 0.58 | 0.0135 | 0.0404 | 0.1211 |
| 2 | 0.0468 | 0.0232 | 0.50 | 0.0175 | 0.0437 | 0.1048 |
| 12 | 0.0425 | 0.0252 | 0.59 | 0.0109 | 0.0382 | 0.1036 |
| 17 | 0.0318 | 0.0237 | 0.74 | 0.0060 | 0.0240 | 0.0960 |
| 11 | 0.0267 | 0.0200 | 0.75 | 0.0049 | 0.0197 | 0.0789 |
| 14 | 0.0256 | 0.0191 | 0.75 | 0.0049 | 0.0198 | 0.0741 |
| $\lambda_0$ | 0.0287 | 0.0068 | 0.24 | 0.0182 | 0.0278 | 0.0444 |
| σ | 1.3828 | 0.1678 | 0.12 | 1.0953 | 1.3678 | 1.7511 |

**Table 4. Upper bold line: best selected models in multimodel GLM with the wildcat density as a response variable and olive crop cover and precipitation as predictors.** We show the AICc values, ΔAICc and Akaike weights of each supported model. Lower bold line: model-averaged coefficients from the multimodel GLM with wildcat density as a response variable and olive crop cover and precipitation as predictors. We show parameter estimates and their standard errors, and the Z values.

| Model | AICc | ΔAICc | Akaike weight |
|---|---|---|---|
| Olive crop cover | -42.68 | | 0.68 |
| Olive crop cover+precipitation | -41.13 | 1.54 | 0.32 |
| **Predictor** | **Estimate** | **SE estimate** | **Z value** |
| Intercept | 0.043 | 0.032 | 1.28 |
| Olive crop cover | 0.010 | 0.0027 | 3.47 |
| Precipitation | 0.00002 | 0.00004 | 0.50 |

Sierra Morena, close to four hundred in Sierras Béticas and less than a dozen in Doñana (see details in Table 5).

## Protected areas cover

The estimated population of wildcats under spatial protection by NPAs is shown in Table 5. Our SDM shows that the potential area of the European wildcat in Andalusia includes 23 of 24

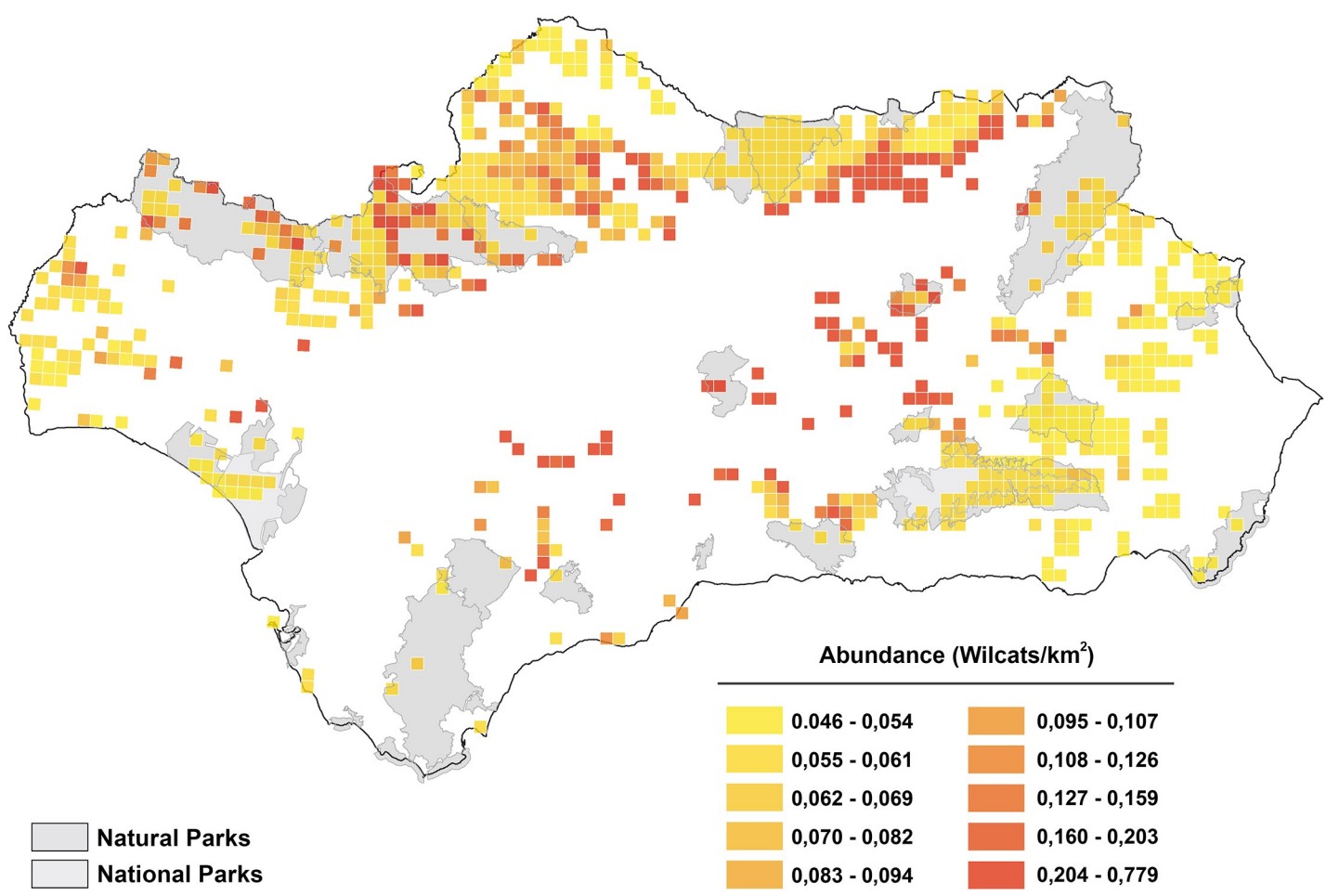

**Abundance (Wilcats/km²)**

| | |
|---|---|
| 0.046 - 0.054 | 0,095 - 0,107 |
| 0,055 - 0,061 | 0,108 - 0,126 |
| 0,062 - 0,069 | 0,127 - 0,159 |
| 0,070 - 0,082 | 0,160 - 0,203 |
| 0,083 - 0,094 | 0,204 - 0,779 |

Natural Parks

National Parks

**Fig 3. GLM-estimated density of the European wildcat in the UTM 5x5 squares with presence predicted by the MaxEnt model.**

**Table 5. Wildcat population estimations in Andalusia, and percentages of the population under spatial protection by national and natural parks (*n*: number of 5x5 km squares).**

| | Wildcat total area (km²) | Wildcat protected area (km²) | $D_{total}$ (indiv./100km²) 95% IC | $D_{protected}$ (indiv./100km²) 95% IC | $N_{total}$ 95%IC | $N_{protegida}$ 95%IC | % protected $N$ |
|---|---|---|---|---|---|---|---|
| Andalusia | 7563.33 | 2530.78 | 11.37 10.50–12.24 *n* = 793 | 8.81 7.76–9.87 *n* = 237 | 860 794–926 | 223 196–250 | 25.9 |
| Sierra Morena range | 4479.62 | 1296.86 | 11.65 10.54–12.76 *n* = 448 | 11.31 8.67–11.94 *n* = 100 | 522 472–571 | 146 112–155 | 27.9 |
| Sierras Béticas range | 2850.07 | 1060.99 | 11.18 9.73–12.63 *n* = 323 | 6.60 5.25–7.95 n = 115 | 319 277–360 | 70 56–84 | 21.9 |
| Doñana range | 233.64 | 172.92 | 8.49 4.02–12.96 *n* = 22 | 5.35 5.31–5.40 n = 13 | 20 9–30 | 9 9–9 | 45 |

Natural Parks and the two existing National Parks (Fig 2). Despite this, only 33.46% of the potential area is protected, and only eight Natural Parks and one National Park have more than one-third of its total area covered by potential wildcat areas. Overall, 25.9% of the estimated wildcats would be under protection: 16.9% in Sierra Morena, 8.1% in the Sierras Béticas and 1.0% in Doñana, out of the total estimated population.

## Discussion

### Camera-trapping for large-scale surveys

Our study represents the first large-scale estimation for any European wildcat population, based both on systematic field surveys and analytical approaches applied on SDMs. For this purpose, the use of camera-trapping has proven to be a logistically viable method for these types of surveys designed for the target species, showing: (1) its utility in situations of very low density (see also the case of the tiger *Panthera tigris* [75, 76]), and (2) its utility for practitioners performing large-scale and long-term monitoring schemes (see the case of the Iberian lynx [80]). These are key advantages over intrusive surveys like radio-tracking, which was previously used for modelling the habitat and distribution of the wildcat in central Europe [51]. Radio-tracking is usually biased towards areas that maximize the chances for captures of individuals to be tagged, and thus is carried out at *a priori* known areas of good density, as revealed by studies carried out on European wildcats [11, 15, 51]. Our camera-trapping survey has allowed for more randomly distributed surveys, hence covering a wider range of density situations. On the other hand, this non-intrusive survey may offer larger sample sizes in the sense of the number of "captured" individuals, preventing redundancy of data. However, in contrast, camera-trapping offers much less data for each individual and cannot allow accurate estimates of home range areas, movements and spatial use of all elements of the landscape. Moreover, for the wildcat, it may present some limitations for correctly identifying individuals, particularly to determine if they are hybrids. Nonetheless, recent studies show that there is a great concordance between external physical features (such as coat pattern) and genetic identity of wildcats, allowing for reliable identifications [30]. Indeed, the presence of cryptic hybrids is very low in the European wildcat populations, <10% [31, 81]. In any case, we recognise that molecular sampling is a necessary tool to obtain the most precise information on the genetic introgression of the domestic cat. Anyway, camera-trapping represents an optimal method to

survey the demographic situation of domestic cats in the wild [47], as the main source of inter-breeding risk.

In fact, camera-trapping is useful for density estimations, a widely acknowledged advantage over other methods [82]. This is true for scat and hair samplings for molecular identification applied to the wildcat as well [42, 46]. However, neither method, especially scat surveys, were useful in our study area. This result could point to severe limitations of scat surveys for European wildcats (see, however, Lozano et al. [83]), again indicating the utility of camera-trapping surveys over large areas.

### Distribution and abundance modelling at a large scale

Our results show the reliability of SDMs to infer the distribution and abundance of an endangered and elusive carnivore. Interestingly, we found that environmental variables can impact distribution and abundance in largely different ways (see next section), supporting the use of our hierarchical approach, beginning with a MaxEnt-based distribution map and then applying a GLM-based abundance analysis on the prior distribution map. This analytical procedure improved the accuracy of results: e.g. if only the abundance model is applied on the whole Andalusia region, then a largely unreliable map is obtained since olive trees are one of the main crops in the Guadalquivir Valley, where wildcats are absent (see Study Area). For our study case, MaxEnt allowed us to perform models using presence-only data, since this algorithm can show a solid performance with small data sets [65, 84], as may be the case in the majority of studies on elusive and rare species. Our distribution model shows a high predictive ability following AUC, suggesting that even in limited sample size scenarios, modelling based on presence data was useful to study wildcat distributions at broad scales. However, we recognize that our results must be taken with some caution. First, the radio-tracking data suggest that the MaxEnt output was conservative. Secondly, environmental covariates and individual behaviour responses (e.g. related to baiting strategies) could affect detection probability [64]. Regardless, we were very cautionary and consistent with the sampling design. In fact, both SCR estimations and the relationship of $D$ with relative abundance shows that the positive records were strongly related to the real abundance of wildcats.

The study of abundance patterns at a regional scale based on GLM models also showed robust results in the case of wildcat in Andalusia. A weakness of our approach was that the smaller optimal patches were penalized, whereas the spatial association of some of them could result in more potential territories than predicted. In any case, this was a marginal situation (see Fig 2) that could be assumed to be of negligible impact at our broad scale, although it should be considered in more local studies.

### Habitat inferences

It is well known that the European wildcat is associated with forests in central Europe [51] and scrublands in Mediterranean landscapes of the Iberian Peninsula [7, 11, 13, 14, 15]. For the Andalusia region, the crops had a key (and negative) effect on the predicted wildcat presence, as it is a landscape feature dictating the significant fragmentation observed in the wildcat distribution. On the other hand, Mediterranean oak forests (mainly made up of holm oaks) had a key positive effect on the predicted wildcat presence, as a habitat that represents one of the main natural landscapes of the Iberian Peninsula [56]. Andalusian wildcats trend to inhabit patches of oak forests (especially xeric ones) somewhat separated from villages and water courses. The positive selection of the oak forest is not supported by some previous studies on wildcat habitat selection in Mediterranean landscapes, which locally described scrublands as key habitats but not oak forests ([7, 11, 13] but see Oliveira et al. [15]). Nonetheless, the

Mediterranean scrublands show a huge diversity and geographic variability. In our study area, most scrubland types in the southern middle of the Iberian Peninsula were available [56]. However, although they may be well represented in the wildcat habitat in Andalusia [14], scrublands had marginal effects in the best predictive model of presence. This could be a result of pooling this complex vegetation into only two types (Table 2), which may not allow differentiation between some types of scrubs selected by wildcats [11, 13] from others avoided by wildcats, such as the hyper-xeric formations of the eastern Andalusian sub-deserts (present study). Scrub–pasture mosaics had a positive effect on wildcats in central Spain [7, 13], and distance to meadows was a key variable for the wildcat prediction models carried out in central Europe [51], showing some positive effects of pasturelands, which we did not find in our study area. This is likely due to the xeric conditions of most of southern Spain.

Regarding wildcat abundance, the only two variables selected by the best model (% of olive tree crops and accumulated annual rainfall) could be related to prey availability. This was confirmed for the case of the olive tree crops, which, although having a lower presence in the habitat of wildcats, they showed a positive relationship to rabbit abundance (see as well Martín-Díaz et al. [14]). During the field walking surveys, we observed that the greater abundances of rabbits were usually associated with mosaics of oak forest/scrublands together with olive tree crops. We detected a negative effect of rainfall on wildcat abundance, being the main reason for the low estimated densities within two of the largest and best conserved patches of Mediterranean forests in Spain: the Alcornocales and Sierra de Grazalema Natural Parks at the western Sierras Béticas, and the Sierra de Cazorla, Segura y las Villas Natural Park at the eastern Sierras Béticas (compare Fig 1 and Fig 2). These two protected areas have the most precipitation for the entire region, along with local areas of western Sierra Morena [56]. The abundance of rabbits is an important variable for the habitat selection models carried out in the Iberian Peninsula [10, 11, 13], where this lagomorph is a key prey for the species [9]. However, for our large-scale survey we found that areas with low rabbit availability were not an exception within the wildcat range (Table 1). Since rodents become a key prey group where rabbits are scarce in southern Spain [8, 85], research on abundance of these preys and its relationships to landscape features is needed to explain the observed negative effect of rainfall.

## Implications for conservation and management

Our results show that the distribution range of one of the largest populations of the European wildcat is actually lower than previously assumed (compare Fig 1 and Fig 2). Moreover, the overall average estimated density in the 19 sampling blocks with confirmed wildcats, 0.069 ±0.0019 wildcats/km$^2$, could be evaluated as very low compared to densities reported by camera-trapping elsewhere: 0.28 ±0.1 wildcats/ km$^2$ in Sicilia [45] and 0.22 ±0.06 wildcats / km$^2$ in Turkey [86]. The protected areas network seems to be insufficient to cover a significant part of the population or a viable nucleus for short-term conservation in genetic terms (with effective population size $N_e$ >50 individuals [87]). Indeed, the most important areas are unprotected (Fig 3). Most of the distribution of wildcats is under hunting estates and it is known that the species severely suffers from illegal and legal control of carnivore mammals to protect lesser game [28]. Moreover, large game hunting can produce indirect negative effects on wildcats by reducing the prey base [10]. The situation is worse for the Sierras Béticas, where the predicted distribution range is both more restricted and fragmented, and probably unconnected to the large and continuous population of Sierra Morena. On the other hand, the whole population seems to be little affected by the hybridisation problem (except in Doñana National Park), which was previously reported by local studies [30, 47]. We found very few domestic cats in the sampled blocks, supporting at a large scale the hypothesis that severe ecological barriers

may be preventing genetic introgression in Mediterranean mountain ranges [47, 88, 89]. In fact, genetic surveys at the Iberian scale have shown very low levels of domestic cat introgression (see e.g. Oliveira, et al. [23]). Nevertheless, in other sites in the Iberian Peninsula, near farms and villages, a relevant presence of domestic cats has been detected [12, 90].

Taking into account this large-scale diagnosis, we have three major recommendations for the studied population: (1) improve monitoring programs of hunting states at least in the main populations; (2) study and improve the connectivity between Sierra Morena and Sierras Béticas, paying special attention to the internal connectivity within the largely fragmented Sierras Béticas population; (3) review the protection laws, since in Andalusia the wildcat is not listed in any threatened category [91], but its present situation does not appear optimistic: <1000 individuals with $N_e$ <100 individuals (10% of $N$ [92]) distributed in a very fragmented range. A similar scenario is highly likely for the rest of Spain, where this species is also not included in any threat category [93].

Our approach can be used not only to update information about the European wildcat at large spatial scales, but also to design viable long-term monitoring programs. Both are actions that have yet to be implemented for such endangered European taxa, or for other felines worldwide. The methodological scheme presented and evaluated here for the wildcat can be useful to better design the limits of protected areas elsewhere.

## Supporting information

**S1 Fig. Output of MaxEnt: ROC curve.**
(DOCX)

**S2 Fig. Output of MaxEnt: Responses of the environmental variables.**
(DOCX)

**S3 Fig. Output of MaxEnt: Jackknife test.**
(DOCX)

## Acknowledgments

We thank Elena Bertos, Elena Ballesteros, Marcos Moleón, Rogelio López, Jesús Bautista, María D. Martínez, Carolina Porto, Carmen Jiménez, Antonio Martín and Antonio Pozo for their valuable help with field work. Juan M. Sáez, Rafael Arenas, Javier Rodríguez, Maribel García, Ángel Arredondo, Miguel A. Díaz-Portero and Emilio González provided information on wildcats in some areas of Sierra Morena and Almería Province.

## Author Contributions

**Conceptualization:** Jose María Gil-Sánchez.

**Data curation:** Jose María Gil-Sánchez, F. Javier Herrera-Sánchez.

**Formal analysis:** Jose María Gil-Sánchez, Jose Miguel Barea-Azcón, José Jiménez, Emilio Virgós.

**Investigation:** Jose María Gil-Sánchez, Jose Miguel Barea-Azcón, Javier Jaramillo, F. Javier Herrera-Sánchez.

**Methodology:** Jose María Gil-Sánchez, Emilio Virgós.

**Resources:** Javier Jaramillo, F. Javier Herrera-Sánchez.

**Software:** Jose Miguel Barea-Azcón, José Jiménez.

**Supervision:** José Jiménez, Emilio Virgós.

**Validation:** Jose Miguel Barea-Azcón, F. Javier Herrera-Sánchez, José Jiménez, Emilio Virgós.

**Visualization:** Jose Miguel Barea-Azcón, Javier Jaramillo, F. Javier Herrera-Sánchez, José Jiménez, Emilio Virgós.

**Writing – original draft:** Jose María Gil-Sánchez.

**Writing – review & editing:** Jose María Gil-Sánchez.

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
