## [Decision Letter · Decision Letter 0]

8 Nov 2019

PONE-D-19-27725

Fragmentation and low density as major conservation challenges for the southernmost populations of the European wildcat

PLOS ONE

Dear Dr. Gil-Sánchez,

Thank you for submitting your manuscript to PLOS ONE. After careful consideration, we feel that it has merit but does not fully meet PLOS ONE’s publication criteria as it currently stands. Therefore, we invite you to submit a revised version of the manuscript that addresses the points raised during the review process.

We would appreciate receiving your revised manuscript by Dec 23 2019 11:59PM. To enhance the reproducibility of your results, we recommend that if applicable you deposit your laboratory protocols in protocols.io, where a protocol can be assigned its own identifier (DOI) such that it can be cited independently in the future. For instructions see: http://journals.plos.org/plosone/s/submission-guidelines#loc-laboratory-protocols

We look forward to receiving your revised manuscript.

Kind regards,

Bi-Song Yue, Ph.D

Academic Editor

PLOS ONE

Journal frequirements;

1. In your Methods section, please include a comment about the state of the pigeons used as live bait following this research. Were they euthanized or housed for use in further research? If any animals were sacrificed by the authors, please include the method of euthanasia and describe any efforts that were undertaken to reduce animal suffering.

2. In your Methods section, please provide additional location information of the study sites, including geographic coordinates for the data set if available.

3. Thank you for including the following funding information within your acknowledgements section; "The research was partially funded by the Consejería de Medio Ambiente y Ordenación del Territorio through the European Union (FEDER Project)  and is part of the Global Change Observatory of Sierra Nevada. J.M.G.-S. was supported by a Prometeo fellowship from the SENESCYT and the national agency for Education and Science of the Government of Ecuador. "

"The author(s) received no specific funding for this work"

Reviewers' comments:

Reviewer's Responses to Questions

**Comments to the Author**

1. Is the manuscript technically sound, and do the data support the conclusions?

Reviewer #1: Partly

Reviewer #2: Yes

2. Has the statistical analysis been performed appropriately and rigorously? 

Reviewer #1: No

Reviewer #2: Yes

3. Have the authors made all data underlying the findings in their manuscript fully available?

Reviewer #1: Yes

Reviewer #2: Yes

4. Is the manuscript presented in an intelligible fashion and written in standard English?

Reviewer #1: No

Reviewer #2: Yes

5. Review Comments to the Author

Reviewer #1: Andalusian wildcats

This is an important paper documenting a rare and cryptic threatened species that involves a huge amount of field work. It is told badly. The authors make the mistake of using Maxent, a programme that sucks the very soul from scientific results like dementers from the world of Harry Potter. Pages of complex analyses do not substitute for some simple mapping and some thought about how to present the key results.

The authors write (the numbers are mine)

1. Our results show that the distribution range is smaller and more highly fragmented than previously assumed.

2. The overall estimated density was very low (0.069 ±0.0019 wildcats/km 2 )

3. and the protected areas network seems to be insufficient to cover a significant part of the population or a viable nucleus in demographic and genetic terms.

4. Indeed, the most important areas remain unprotected.

5. Our main recommendations are to improve the protected area network and/or vigilance programs in hunting estates,

6. in addition to studying and improving connectivity between the main population patches.

Those are interesting results, but it’s far from clear how one goes from figure 1 to figure 2, to figure 3 and these conclusions.

For 1, the authors would need to document previous estimates and previous assertions that the range is continuous. The would also need to convince the reader that the range is fragmented. A quick look at Google Earth reminded me of what I remembered about this area: much of the land below 200m elevations is extensively converted to agriculture.

Doñana, is the important exception. To address (2), it has a low density.

The largest block of potential habitat is Sierra Morena. The authors models suggest the species is widespread there on the basis of intensive sample in the east, (points 1 to 9) and just one sample in the west (10). Some of the highest densities occur there — >0.2 cats per km2, but these predictions are outside the area of systematic sampling. I’d need this to be explained.

This is where Maxent so often fails. The authors fall back on its black magic, when simple GIS mapping would be so much more important. The model predicts that the cat is missing from large areas of this Sierra’s national parks. Why? What’s wrong with the habitat? And how might one map that?

The southern distribution clusters into a southwestern area (points 21-23) and the eastern one that I will call the Sierra Nevada.

The southwestern one predicts very little habitat and estimates very low densities. Why?

The Sierrra Nevadas are well sampled (points 11-20), predicted to be habitat (figure 2). The densities are predicted to be low.

In addition to these core areas is a substantial scatter of small, isolated patches with predicted very high densities of cats (>0.2), running north of points 22, 20 to 18. These are areas that weren’t surveyed. How do we know there are cats there and how to we know what the densities are? Their existence seems to be the key result in justifying the conclusion about fragmentation.

It also seems to be critical in point 4, since Doñana, Morena, Sierra Nevada, and the southwest do have extensive protected areas. The authors have not convinced me that that there are many cats outside these areas.

Finally, points 5 and 6 need specifics. Where exactly would put new protected areas? And where are the hunting estates that one might wish to influence?

In sum, I trust the authors instincts and experience in the conclusions they present. But they should start from them and work backward to justify each one with the simplest analyses possible.

Reviewer #2: Overall this is an excellent paper. The authors SDM approach to this question is robust. Some of their English phrasing needs a bit of work. However, addition of genetic data would make this a much stronger paper and so it is not clear to me why the authors did not collect tissue or blood samples from at least the 9 radio-tagged animals for genetic analysis? And why hair snares were not put out at the camera locations? Even though they state that “Three types of non-intrusive field surveys have been successfully developed for the wildcat: camera-trapping, scat sampling for molecular identification and hair sampling, also for genetics. We used the first two methods, while ongoing field studies by our team are showing a lesser efficiency of hair traps in our study area.” They do not report on any genetic analyses - even if the hair snares are not very efficient they could provide at least some samples for genetic analysis. Also it is curious why they did not put out live traps after identifying animals with the camera traps in order to try and capture these individuals – seems an opportunity lost. As a result they rely on visual determination of hybrids which is sketchy at best. They have no estimates of effective population size or genetic variation and they do not identify or at least to not report numbers of males and females. Therefore they can estimate densities but and population size but this doesn’t translate into how viable these populations are. If they have any genetic data they should include it. Despite the lack of genetic data this is still a very valuable contribution to the literature on European wild cats.

Lines 31-32 numerous grammatical mistakes in the first sentence

“On” should be “of”

Delete the in “the population size”

Population dynamics not just dynamics

Conservation actions not conservation measures

Line 32 delete “huge numbers of examples” and add many examples

Line 49 “… and the protected areas network seems to be insufficient to cover a significant part of the population or a viable nucleus in demographic and genetic terms.” You don’t have any genetic data so this is pure conjecture.

Lines 57-58 see comments above

Line 116 hybridization not hybridization

Line 122 capitalize the f in fig.1

Line 132 add “and is densely populated by humans” after cereal crops and delete the next sentence.

Line 133 add “in this area” after wild landscape

Line 163 delete “they are” in front of representative

Line 200 what does “conceptual groups” mean?

Line 251 capitalize fig.

6. PLOS authors have the option to publish the peer review history of their article (what does this mean?). If published, this will include your full peer review and any attached files.

Reviewer #1: No

Reviewer #2: No

---

## [Author Response · Author response to Decision Letter 0]

10 Dec 2019

Response to Reviewers

Journal requirements

1. In your Methods section, please include a comment about the state of the pigeons used as live bait following this research. Were they euthanized or housed for use in further research? If any animals were sacrificed by the authors, please include the method of euthanasia and describe any efforts that were undertaken to reduce animal suffering.

** Our response: done in L265-266 of “Revised ms”.

2. In your Methods section, please provide additional location information of the study sites, including geographic coordinates for the data set if available.

** Our response: done in second column of Table 1 of “Revised ms”.

3. Thank you for including the following funding information within your acknowledgements section; "The research was partially funded by the Consejería de Medio Ambiente y Ordenación del Territorio through the European Union (FEDER Project) and is part of the Global Change Observatory of Sierra Nevada. J.M.G.-S. was supported by a Prometeo fellowship from the SENESCYT and the national agency for Education and Science of the Government of Ecuador. "

"The author(s) received no specific funding for this work"

** Our response: done, deleted in the “Revised ms”.

Review Comments to the Author

Reviewer #1: Andalusian wildcats

This is an important paper documenting a rare and cryptic threatened species that involves a huge amount of field work. It is told badly. The authors make the mistake of using Maxent, a programme that sucks the very soul from scientific results like dementers from the world of Harry Potter. Pages of complex analyses do not substitute for some simple mapping and some thought about how to present the key results.

** Our response: We acknowledge the reviewer for thinking that our work is an important paper. However, as we will discuss below, MaxEnt is one of the most prominent methods for species distribution modelling, and it has been proved robust and efficient compared to other statistical methods (GLM, GAM, GARP, Random Forest or Mahalanobis distance). Moreover, MaxEnt appears as key statistical tool in several articles recently published in PLoSONe, please, see some examples listed below, including an article with a feline study case:

Wiese D, Escalante AA, Murphy H, Henry KA, Gutierrez-Velez VH (2019) Integrating environmental and neighborhood factors in MaxEnt modeling to predict species distributions: A case study of Aedes albopictus in southeastern Pennsylvania. PLoS ONE 14(10): e0223821. https://doi.org/10.1371/journal.pone.0223821

Angelieri CCS, Adams-Hosking C, Ferraz KMPMdB, de Souza MP, McAlpine CA (2016) Using Species Distribution Models to Predict Potential Landscape Restoration Effects on Puma Conservation. PLoS ONE 11(1): e0145232. https://doi.org/10.1371/journal.pone.0145232

Barnhart PR, Gillam EH (2016) Understanding Peripheral Bat Populations Using Maximum-Entropy Suitability Modeling. PLoS ONE 11(12): e0152508. https://doi.org/10.1371/journal.pone.0152508

Fourcade Y, Engler JO, Rödder D, Secondi J (2014) Mapping Species Distributions with MAXENT Using a Geographically Biased Sample of Presence Data: A Performance Assessment of Methods for Correcting Sampling Bias. PLoS ONE 9(5): e97122. https://doi.org/10.1371/journal.pone.0097122

The authors write (the numbers are mine)

1. Our results show that the distribution range is smaller and more highly fragmented than previously assumed.

2. The overall estimated density was very low (0.069 ±0.0019 wildcats/km 2 )

3. and the protected areas network seems to be insufficient to cover a significant part of the population or a viable nucleus in demographic and genetic terms.

4. Indeed, the most important areas remain unprotected.

5. Our main recommendations are to improve the protected area network and/or vigilance programs in hunting estates,

6. in addition to studying and improving connectivity between the main population patches.

Those are interesting results, but it’s far from clear how one goes from figure 1 to figure 2, to figure 3 and these conclusions.

For 1, the authors would need to document previous estimates and previous assertions that the range is continuous. The would also need to convince the reader that the range is fragmented. A quick look at Google Earth reminded me of what I remembered about this area: much of the land below 200m elevations is extensively converted to agriculture. Doñana, is the important exception. 

** Our response: We exactly wrote: (L558-L559 in the first version) “Our results show that the distribution range of one of the largest populations of the European wildcat is actually lower than previously assumed.”

We acknowledge that a reference in the end of this sentence is necessary (ref. [92]) and a call to Fig 1, where we present the IUCN map for the European wildcat. We have added “(compare Fig. 1 and Fig. 2)” in L568-L569 of of “Revised ms”.

To address (2), it has a low density. The largest block of potential habitat is Sierra Morena. The authors models suggest the species is widespread there on the basis of intensive sample in the east, (points 1 to 9) and just one sample in the west (10). Some of the highest densities occur there — >0.2 cats per km2, but these predictions are outside the area of systematic sampling. I’d need this to be explained.

** Our response: we are using SDMs since it is not viable to conduct a complete field survey covering the whole Andalusian region. Therefore, please note that we did not carry out a systematic sampling. In the case of Sierra Morena, this is a homogeneous range (as we expose in “Study Area” section), but in any case our sampling was designed to cover most of it low landscape variability, as we expose in “Field Surveys” section:

L160-L166 in the first version: Twenty-two survey blocks (Fig 1) were distributed in Sierra Morena (10 blocks) and Sierras Béticas (12 blocks) between 2011 and 2015; this distribution was biased towards eastern Andalusia mainly because the habitats there are much more heterogeneous and they are representative of the other areas. All sampling blocks were selected first assuming that they were included in the potential habitats of the wildcat (see study area), and second, attempting to represent the main forest and scrubland types of the overall potential range within Andalusia.

This is where Maxent so often fails. The authors fall back on its black magic, when simple GIS mapping would be so much more important. The model predicts that the cat is missing from large areas of this Sierra’s national parks. Why? What’s wrong with the habitat? And how might one map that?

** Our response: We used MaXent as a tool to produce a spatial prediction of relative suitability of the different regions of Andalusia. The function produced by MaXent was used to predict how suitable one area is for wildcats. Then, as any other predictive mathematical tool, this function can be used to infer if wildcats can be present or not over different spatial points. To calculate number of potential wildcats in each suitable area we used the equation yield by a GLM performed with a subset of locations where wildcat density was known by means of SCR methods (see also Tôrres, N. M., De Marco, P., Santos, T., Silveira, L., de Almeida Jácomo, A. T., & Diniz‐Filho, J. A. (2012). Can species distribution modelling provide estimates of population densities? A case study with jaguars in the Neotropics. Diversity and Distributions, 18(6), 615-627). Then, our estimates are the product of a step by step process where common analytical tools such as MaxEnt (for presence only modelling) and GLM (used with density data obtained for SCR methods) were used to estimate number of animals across the space. Indeed, this a very common methodology in conservation science, where predictive modelling has been used to cope with different conservation problems (translocations, efficient sampling of elusive species, effects of climate change or invasive species and some more. e.g. Rodríguez, J. P., Brotons, L., Bustamante, J., & Seoane, J. (2007). The application of predictive modelling of species distribution to biodiversity conservation. Diversity and Distributions, 13(3), 243-251;Guisan, A., Tingley, R., Baumgartner, J. B., Naujokaitis‐Lewis, I., Sutcliffe, P. R., Tulloch, A. I., ... & Martin, T. G. (2013). Predicting species distributions for conservation decisions. Ecology letters, 16(12), 1424-1435). This approach is much better than simple mapping of presence as suggested by referee. Simple mapping did not allows for extrapolation of other potential overlooked populations. It is important to remember that atlas dataset (the usual way to present a species distribution map) is very constrained by where volunteers are present, and even some large populations can be then be overlooked. We understand some people is not very happy with predictive modelling in conservation, but is one of the most usual and powerful analytical tools in modern Conservation Biology (see references above). Another question is if MaXent is the best option to produce a suitability or occurrence map. However, despite criticisms MaXent is one of the most prominent methods for analyzing presence-only data, and it has been proved robust and efficient compared to other statistical methods (GLM, GAM, GARP, Random Forest or Mahalanobis distance, e.g. Elith, J.,Graham, C.H.,Anderson, R.P.etal.(2006).Novel methods improve prediction of species’ distributions from occurrence data.Ecography,29,129–151.;Elith, J., Phillips, S. J., Hastie, T., Dudík, M., Chee, Y. E., & Yates, C. J. (2011). A statistical explanation of MaxEnt for ecologists.Diversity and distributions, 17(1), 43-57). For these reasons, we are confident about the usefulness of our approach and statistical tools to predict best areas for wildcats and potential numbers at different regions of Andalusia. Moreover, please, keep in mind that we have gone further by carrying out an independent validation of our MaxEnt maps through radio-tracking data.

The southern distribution clusters into a southwestern area (points 21-23) and the eastern one that I will call the Sierra Nevada. The southwestern one predicts very little habitat and estimates very low densities. Why?

** Our response: Please, go to lines 542-546 of Discussion in the first version: “We detected a negative effect of rainfall on wildcat abundance, being the main reason for the low estimated densities within two of the largest and best conserved patches of Mediterranean forests in Spain: the Alcornocales and Sierra de Grazalema Natural Parks at the western Sierras Béticas”. We think it is well explained here. Moreover, we have consulted to several field biologists and naturalists working within this area and nobody gave us one confirmed record of wildcat. We deleted this sentence in an early version to reduce our ms.

The Sierrra Nevadas are well sampled (points 11-20), predicted to be habitat (figure 2). The densities are predicted to be low. In addition to these core areas is a substantial scatter of small, isolated patches with predicted very high densities of cats (>0.2), running north of points 22, 20 to 18. These are areas that weren’t surveyed. How do we know there are cats there and how to we know what the densities are? Their existence seems to be the key result in justifying the conclusion about fragmentation.

** Our response: we know the presence of wild cats in some of this patches (see the withe dot in between, Fig. 1; and from other data not considered in this ms, as road kills, poaching, sightings…). In any case, this small and scattered patches represent a very marginal contribution to the Betic subpopulation, and huge fragmentation affects to eastern Betic Mountains range, where the main meta-population of the Betic Mountains survives. Thus, the very small patches running north of points 22, 20 to 18 are not really a key result in justifying the conclusion about fragmentation.

It also seems to be critical in point 4, since Doñana, Morena, Sierra Nevada, and the southwest do have extensive protected areas. The authors have not convinced me that that there are many cats outside these areas.

** Our response: Please, see Table 5. Sorry, it is not a matter of opinion.

Finally, points 5 and 6 need specifics. Where exactly would put new protected areas? And where are the hunting estates that one might wish to influence?

** Our response: On the reviewer’s first question, the creation of new protected areas (or increasing the current ones) must be solved by the Environmental authorities. Our study obviously is of great value to them, it is one of our practical application aims for conservation in practice in a short time (it is really urgent for Mediterranean wildcats!!!), but our specific goals fall out this key action, which requires more in depth research including key sociological aspects. On the second question, we provide a map (fig .3) that helps the Environmental authorities to detect the priority hunting estates that one might wish to influence.

In sum, I trust the authors instincts and experience in the conclusions they present. But they should start from them and work backward to justify each one with the simplest analyses possible.

** Our response: We hope that our explanations correctly solve the doubts raised by the reviewer.

 

Reviewer #2: Overall this is an excellent paper. The authors SDM approach to this question is robust. 

** Our response: We sincerely acknowledge the reviewer for thinking that our work is an excellent paper.

Some of their English phrasing needs a bit of work. 

** Our response: we have added all the corrections provided below by rev.2. The English was revised by a professional. 

However, addition of genetic data would make this a much stronger paper and so it is not clear to me why the authors did not collect tissue or blood samples from at least the 9 radio-tagged animals for genetic analysis? 

** Our response: actually we collected samples of all of them, and some of these individuals were genetically analyzed, please see ref [30], where 17 wildcats of our study area genetically analyzed were examined. Indeed, these wildcat sample has been recently analyzed using SNPs, confirming no domestic cat introgression (Mattucci, F., Galaverni, M., Lyons, L.A. et al. Genomic approaches to identify hybrids and estimate admixture times in European wildcat populations. Sci Rep 9, 11612 (2019) doi:10.1038/s41598-019-48002-w). We have added this in the new version (L198-L199).

And why hair snares were not put out at the camera locations? Even though they state that “Three types of non-intrusive field surveys have been successfully developed for the wildcat: camera-trapping, scat sampling for molecular identification and hair sampling, also for genetics. We used the first two methods, while ongoing field studies by our team are showing a lesser efficiency of hair traps in our study area.” They do not report on any genetic analyses - even if the hair snares are not very efficient they could provide at least some samples for genetic analysis. 

** Our response: please, see previous response. And, on the other hand, we were not able to collect hair samples since our wildcat rarely rubbed in our baits, which were checked with camera traps.

Also it is curious why they did not put out live traps after identifying animals with the camera traps in order to try and capture these individuals – seems an opportunity lost. 

** Our response: it was logistically inviable in our large-scale survey. We only could capture wildcats in block 18, as we explain in the text.

As a result they rely on visual determination of hybrids which is sketchy at best. 

** Our response: we detected 50 putative pure individuals (Table 1), following a strict protocol for identification (see ref [30], cited in Methods); the phenotypic traits of genetically pure individuals of our study area are very constant, and we only considered typical wildcats as “pure” (most of them identical to WC1 in fig. 2 of [30]). We did not explain here the identification protocol since it is well explained elsewhere, thus, we put only the references (e.g. references [30] and [47]). Our three detected hybrids where very close to typical wildcats but they had wide white patches in pelage. We have added this in the new version (L199-L204).

They have no estimates of effective population size or genetic variation and they do not identify or at least to not report numbers of males and females. Therefore they can estimate densities but and population size but this doesn’t translate into how viable these populations are. If they have any genetic data they should include it. 

** Our response: Sorry, but it was not our goal to run a PVA.

Despite the lack of genetic data this is still a very valuable contribution to the literature on European wild cats.

** Our response: We are very grateful to the reviewer.

Lines 31-32 numerous grammatical mistakes in the first sentence

“On” should be “of” 

** Our response: done

Delete the in “the population size” 

** Our response: done

Population dynamics not just dynamics 

** Our response: done

Conservation actions not conservation measures 

** Our response: done

Line 32 delete “huge numbers of examples” and add many examples 

** Our response: done

Line 49 “… and the protected areas network seems to be insufficient to cover a significant part of the population or a viable nucleus in demographic and genetic terms.” You don’t have any genetic data so this is pure conjecture. 

** Our response: We have delete “genetic” in the new version.

Lines 57-58 see comments above 

** Our response: done

Line 116 hybridization not hybridization 

** Our response: done

Line 122 capitalize the f in fig.1 

** Our response: done

Line 132 add “and is densely populated by humans” after cereal crops and delete the next sentence.

 ** Our response: done

Line 133 add “in this area” after wild landscape 

** Our response: done

Line 163 delete “they are” in front of representative 

** Our response: done

Line 200 what does “conceptual groups” mean? 

** Our response: climatic, relief, vegetation, water availability and human presence: we have added this in L208-L209 of the new version.

Line 251 capitalize fig. 

** Our response: done

---

## [Editor Report · Decision Letter 1]

27 Dec 2019

Fragmentation and low density as major conservation challenges for the southernmost populations of the European wildcat

PONE-D-19-27725R1

Dear Dr. Gil-Sánchez,

We are pleased to inform you that your manuscript has been judged scientifically suitable for publication and will be formally accepted for publication once it complies with all outstanding technical requirements.

With kind regards,

Bi-Song Yue, Ph.D

Academic Editor

PLOS ONE

---

## [Editor Report · Acceptance letter]

9 Jan 2020

PONE-D-19-27725R1 

Fragmentation and low density as major conservation challenges for the southernmost populations of the European wildcat 

Dear Dr. Gil-Sánchez:

I am pleased to inform you that your manuscript has been deemed suitable for publication in PLOS ONE. Congratulations! Your manuscript is now with our production department. 

With kind regards,

on behalf of

Dr. Bi-Song Yue 

Academic Editor

PLOS ONE